# The Effect of Trunk Position on Attentional Disengagement in Unilateral Spatial Neglect

**Kohei Shida [1,2,\*], Kazu Amimoto [2] , Kazuhiro Fukata [1], Shinpei Osaki [2,3], Hidetoshi Takahashi [1] and Shigeru Makita [1]**

1   Department of Rehabilitation Center, Saitama Medical University International Medical Center, 1397-1, Yamane, Hidaka 350-1298, Japan

2   Department of Physical Therapy, Faculty of Human Health Sciences, Tokyo Metropolitan University, 7-2-10, Higashiogu, Arakawa-ku, Tokyo 116-8551, Japan

3   Department of Rehabilitation, Kansai Electric Power Hospital, 2-1-7, Fukushima, Fukushima-ku, Osaka-shi 553-0003, Japan

\*   Correspondence: koheishida18@gmail.com; Tel.: +81-80-1832-7488

**Abstract:** Unilateral spatial neglect (USN) causes difficulties in disengaging attention from the right side to unexpected targets on the left. However, the relationship between egocentric spatial position and attentional disengagement remains unclear. Therefore, this study aimed to clarify the relationship between trunk position and attentional disengagement. Thirty-eight patients with early stroke onset were classified as follows: USN (n = 18), right brain damage without USN (n = 10), and left brain damage (n = 10). The primary outcome was reaction time (RT) in the modified Posner task (MPT). The MPT comprised a condition in which the preceding cue and target direction were the same (valid condition) and a condition in which the directions were opposite (invalid condition). RT to the target was calculated. The MPT was performed in three different trunk positions (trunk midline, left, and right). In each group, the RT was compared on the basis of the stimulus conditions and trunk position. The RT was delayed in the valid and invalid left conditions, especially in the invalid left condition. The RT of the trunk right condition was significantly reduced compared with that of trunk midline and left conditions in the invalid left condition. Thus, trunk position influences attentional disengagement. This study contributes to the rehabilitation of patients with neglect symptoms.

**Keywords:** unilateral spatial neglect; modified Posner task; trunk position





## 1. Introduction

Unilateral spatial neglect (USN) reduces or eliminates responses to stimuli from the contralesional side of space [1]. USN can occur in both right brain damage (RBD) and left brain damage (LBD) [2,3]. The review by Esposito et al. reported that the occurrence rates of USN were 38% after RBD and 18% after LBD in acute phase and were 20% after RBD and 13% after LBD in chronic phase (>1 year) [2]. It was reported that patients with RBD had both higher severity and chronicity of neglect symptoms compared with patients with LBD [4]. USN is associated with negative clinical outcomes [3,5–7], and the severity of USN in the acute phase negatively affects the long-term functional prognosis and the degree of independence [7].

There are a variety of tests to assess neglect symptoms; primarily, the Behavioral Inattention Test (BIT) and Catherine Bergego Scale (CBS) are used. The BIT is the most widely used battery to assess neglect symptoms [8], and it can assess various types of neglect symptoms, such as visual search, visuospatial perception, and spatial representations. The CBS assesses neglect symptoms in activities of daily living (ADL) on the basis of observation and has reliability and validity [9]. However, discrepancies between the BIT and the CBS can occur; some patients show neglect symptoms while performing ADL, even when the BIT score is above the cut-off [10]. This is due to the fact that the BIT is an assessment tool

that focuses on endogenous attention, suggesting that a detailed assessment of exogenous attention is important.

The Posner task is a computerized reaction time test for neuropsychological assessment, which can provide a detailed evaluation of both endogenous and exogenous attention [11–18]. Endogenous attention implies goal-directed attention by voluntary action, whereas exogenous attention implies stimulus-driven attention by external stimuli. USN is considered a disorder of the attention networks; endogenous and exogenous attention each consist of different attentional networks [19]. The first network is the dorsal attention network, which comprises the superior parietal lobule, intraparietal sulcus, and frontal eye field in the right hemisphere and is associated with endogenous attention. The second network is the ventral attention network, which comprises the temporoparietal junction (TPJ), middle frontal gyri, and inferior frontal gyri and is related to exogenous attention. The Posner test is highly sensitive in detecting USN, not only in acute patients with obvious neglect symptoms but also in chronic patients with mild neglect symptoms [11]. This task can evaluate endogenous attention when the direction of the preceding peripheral cue (arrow direction) at the center of the screen matches that of the target (valid condition), and there is a switch from endogenous to exogenous attention when the direction of the cue and the target are mismatched (invalid condition). The Posner task is useful for assessing the following three neglect symptoms [12]: (a) impairments in visual perception and direction of attention, that is, delayed reaction time or reduced accuracy to targets presented in the left space compared with the right space; (b) impaired spatial attention reorientation (exogeneous attention), that is, delayed response to a target that appears in an unattended location compared with a target that appears in an attended location; and (c) impairment of attentional disengagement (switching from endogenous to exogenous attention), that is, releasing attention from the right space and redirecting it to unexpected targets that appear in the left space. Attentional disengagement can be assessed in the invalid condition of the Posner task. In the invalid condition, the patient is required to disengage attention from the cued stimulus and reorient attention to the new target, which requires higher attentional function than simply responding to a target that has appeared in a single target in the neglected space.

It is necessary to appropriately integrate information from eye, head, and trunk positions to define an egocentric reference frame [20–25]. Both hemispheres encode this information from the contralateral space and interact with each other to maintain balance [26]. In cases of right brain damage, an interhemispheric imbalance leads to a rightward deviation of the egocentric space reference, thus resulting in difficulty with a visual search of the left space [24,25]. It has been reported that the head-on-trunk position is related to not only endogenous visual search but also reaction time for stimuli that appear in the neglected space [23]. Rorden et al. [23] investigated the relationship between trunk position and stimulus-driven attention. The reaction time was reduced when the monitor was placed at +40° to the right of the mid-sagittal trunk position (relative left rotation of the trunk) in USN patients. This suggests that spatial factors centered on the trunk and temporal factors for finding the target are related and that the time required for response depends on the trunk position, even if the positions of the eye and head are the same with respect to the target information. However, despite the relationship between trunk position and exogenous attention being investigated by Rorden et al. [23], it is unclear how trunk position affects the switch from endogenous to exogenous attention (attentional disengagement), as per assessment by the Posner task. Therefore, the present study aimed to clarify the relationship between trunk position and attentional disengagement.

## 2. Materials and Methods

### 2.1. Study Design and Statement of Ethics

This was a prospective cross-sectional study. It was approved by the Research Ethics Committee of Saitama Medical University International Medical Center (Approval No. 09-078) and Tokyo Metropolitan University (Approval no. 20043). Written informed consent

was obtained from all participants and the study was conducted in accordance with the Declaration of Helsinki.

### 2.2. Participants

Patients with a diagnosis of stroke who were admitted to the International Medical Center of Saitama Medical University between March 2020 and March 2022 were considered for participation. Patients with early stroke onset who met the inclusion and exclusion criteria described below were included.

The inclusion criteria were as follows: (a) within 30 days of onset, (b) first onset, (c) age $\geq$ 20 years, and (d) right-handed. The exclusion criteria were (a) sub-tentorial or bilateral lesions, (b) difficulty in understanding simple instructions owing to cognitive dysfunction (Mini-Mental State Examination (MMSE) score >21), and (d) visual deficits (e.g., hemianopsia) assessed by the confrontation visual field testing. On the basis of these criteria, 38 patients (28 patients with RBD and 10 patients with LBD) participated in this study. Patients with a BIT score < 131 points and/or a CBS score $\geq$ 1 point were diagnosed with USN. Eventually, they were classified into three groups as follows: RBD with USN group (USN group, *n* = 18), RBD without USN group (RBD group, *n* = 10), and LBD without USN group (LBD group, *n* = 10).

### 2.3. Clinical Assessment

BIT scoring was performed in both the USN and RBD groups, and CBS scoring was performed in all groups to assess neglect symptoms. The BIT is a paper-and-pencil test that consists of six items: line crossing, letter cancellation, star cancellation, figure and shape copying, line bisection, and representational drawing. Patients with a BIT score of <131 points were classified as having USN. The CBS assesses neglect symptoms in ADL situations on the basis of observation and consists of 10 subscales: grooming, dressing, eating, mouth cleaning, gaze orientation, left limb knowledge, auditory attention, collisions, spatial orientation, and finding personal belongings. Each sub-item was scored from 0 (no neglect) to 3 (severe neglect), for a total score of 30 points. The CBS score was assessed by an occupational therapist or a stroke nurse. Patients with a CBS score $\geq$ 1 point were defined as having USN.

The MMSE was performed to confirm that cognitive function was adequate for the accurate assessment of neglect symptoms. Additionally, age, sex, number of days since onset, and disease type were investigated. The lesion sites of each group were identified by a rehabilitation physician (H.T.).

### 2.4. The Modified Posner Task

The Posner task is a computerized test that assesses visual attention on the basis of the relationship between the direction of the preceding cue and the target's position. In this study, we used the modified Posner task (MPT) developed by Osaki et al. [15], in which targets were presented in four frames: upper left, lower left, upper right, and lower right (Figure 1).

A 17-inch computer display (ROG STRIX GL703VM SCAR Edition, ASUSTeK Computer Inc., Taiwan, China) and software (SuperLab 5.0, Cedrus Corporation, San Pedro, CA, USA) were used for the MPT. A keypad (TK-TCM011, ELECOM Co., Ltd., Osaka, Japan) was used to obtain the response behavior. The patient was seated 50 cm away from the computer during the test. The display comprises a central fixation cross and four square frames located to the right and left along the horizontal meridian. The length of one side of the square frame was 3.8°, relative to the visual angle. The diameter of the circular target appearing in the frames was 2.4°, and it appeared 16.4° away from the fixation cross, relative to the visual angle.

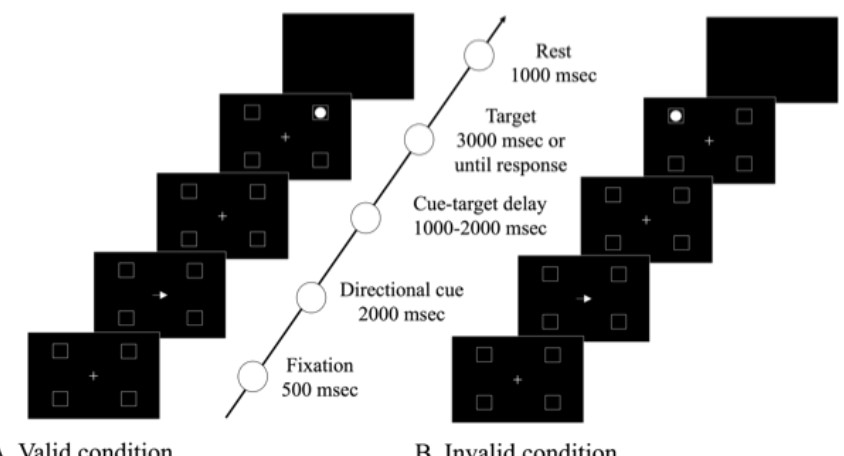

**Figure 1.** Modified Posner task method. (**A**) valid condition, (**B**) invalid condition.

The task starts when the center circle changes from red to green, followed by the appearance of the four frames and fixation cross in the center of the screen. Then, 500 ms later, an arrow pointing either left or right is shown for 2000 ms as a cue. The target appears in one of the four frames 1000–2000 ms after the cue appears. Targets are presented until the patient presses a button or 3000 ms has passed, whichever comes first. The patients are instructed to press the button as quickly as possible when the target is detected. Each session consisted of 60 trials, with 80% (48 trials) of valid conditions in the same direction as the cue and target and 20% (12 trials) of invalid conditions with different directions of the cue and target. The time from target appearance to button pressing by the patient was recorded. Before the MPT was performed, the task was explained to the patient and practice sessions were conducted to ensure that the patient fully understood the method.

To evaluate the effect of the trunk position, the horizontal position of the monitor was set to the midline of the trunk of the patient (0°: trunk midline), 40° to the right (+40°: trunk right), and 40° to the left (−40°: trunk left) (Figure 2). Trunk orientation was kept constant, and head position was rotated relative to the monitor. The coordinates of the retina at which the cues and targets were presented were always kept constant, and only their position relative to the trunk position was manipulated. The MPT was performed once at each monitoring location with a 5 min rest in between, and the order in which the tasks were performed was randomized.

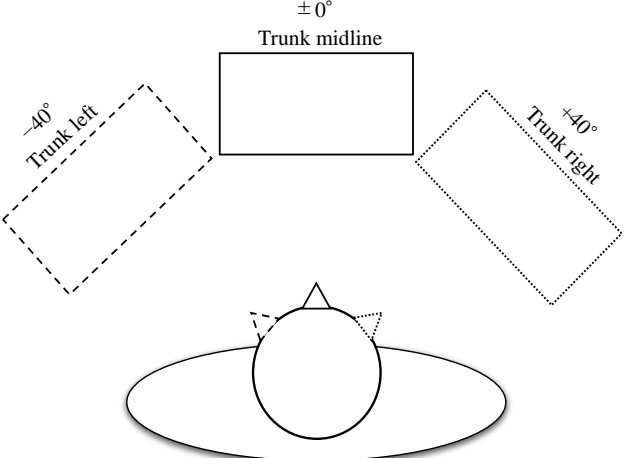

**Figure 2.** Trunk position. The horizontal position of the monitor was pseudo-randomly set to the midline of the subject's trunk (0°: trunk midline), 40° to the right (+40°: trunk right), or 40° to the left (−40°: trunk left).

*2.5. Data Analysis*

Statistical analysis was performed using SPSS statistical software version 26.0 (IBM Corporation, Armonk, NY, USA). One-way analysis of variance (ANOVA) was used to compare age, days since onset, and MMSE scores among the three groups, and multiple comparison tests were performed when there was a significant difference. In addition, the unpaired t-test was used to compare BIT scores between the USN and RBD groups.

In the MPT, the mean reaction time and the response rate were calculated for four stimulus conditions: (a) the condition that the cue pointed to the left and the target appeared in the upper left or lower left (valid left), (b) the condition that the cue pointed to the right and the target appeared in the upper right or lower right (valid right), (c) the condition that the cue pointed to the right and the target appeared in the upper left or lower left (invalid left), and (d) the condition that the cue pointed to the left and the target appeared in the upper right or lower right (invalid right).

A two-way repeated measures ANOVA was used to compare the reaction times for groups (USN, RBD, and LBD) × target conditions (valid left, valid right, invalid left, and invalid right) in the trunk midline condition. A one-way repeated measures ANOVA or the Friedman test was used to compare reaction times and response rates by trunk position for each valid left, valid right, invalid left, and invalid right condition. The reason for using the Friedman test is that the reaction times of valid left in the trunk left condition and response rates of valid and invalid conditions in the USN group did not show normality. In addition, the difference in reaction times was calculated by subtracting the trunk right from the trunk left for each valid left, valid right, invalid left, and invalid right condition.

Difference in reaction times = (reaction times of trunk left) − (reaction times of trunk right).

A one-way repeated ANOVA was used to compare the differences in reaction times among conditions. The post-hoc tests were performed when there were significant differences in the ANOVA (Tukey's HSD) or Friedman tests (Bonferroni correction). The F-value, degree of freedom $(df)_1$, and $df_2$ were calculated for the ANOVA. The t-value and df were calculated for the unpaired t-test. We calculated the $\chi^2$ values and df for the Freidman test. Statistical significance was set at $\alpha$ = 0.05.

## 3. Results

*3.1. Clinical Data*

Demographic and clinical data are shown in Table 1. There were no significant differences in age or time from onset among the three groups. MMSE scores were significantly lower in the USN group than in the RBD group ($F_{(2, 35)}$ = 4.54; $p$ = 0.02). The BIT scores were significantly lower in the USN group than in the RBD group (t = −4.85; df = 26; $p$ = 0.001), and all subitems except for line crossing were significantly lower as well.

**Table 1.** Demographic and clinical data of all participants.

| | USN Group (*n* = 18) | RBD Group (*n* = 10) | LBD Group (*n* = 10) | Group Difference |
|---|---|---|---|---|
| Age (years) | 69.7 ± 11.0 | 61.6 ± 11.0 | 65.9 ± 11.7 | $F_{(2, 35)}$ = 1.609, $p$ = 0.215 |
| Sex (male/female) | 10 (55.6)/8 (44.4) | 8 (80.0)/2 (20.0) | 2 (20.0)/8 (80.0) | |
| Types of disease (infarction/hemorrhage) | 11 (61.1)/7 (38.9) | 4 (40.0)/6 (60.0) | 7 (70.0)/3 (30.0) | |
| Period from onset (days) | 16.1 ± 7.5 | 15.2 ± 5.5 | 15.4 ± 6.9 | $F_{(2, 35)}$ = 0.060, $p$ = 0.942 |
| MMSE (points) | 24.6 ± 2.5 * | 27.6 ± 2.7 * | 26.6 ± 2.4 | $F_{(2, 35)}$ = 4.548, $p$ = 0.018 |
| BIT (points) | 118.2 ± 18.3 * | 140.7 ± 4.1 * | | t = −4.853, df = 26, $p$ = 0.001 |
| Line crossing | 35.7 ± 0.80 | 36.0 ± 0.0 | | t = 1.426, df = 26, $p$ = 0.172 |
| Letter cancellation | 27.3 ± 10.1 * | 36.5 ± 2.3 * | | t = −3.570, df = 26, $p$ = 0.002 |
| Star cancellation | 45.6 ± 8.7 * | 53.6 ± 0.92 * | | t = −3.762, df = 26, $p$ = 0.001 |
| Figure and shape copying | 1.7 ± 1.2 * | 3.4 ± 0.49 * | | t = −4.907, df = 26, $p$ < 0.001 |
| Line bisection | 6.6 ± 2.1 * | 8.4 ± 1.3 * | | t = −2.449, df = 26, $p$ = 0.021 |

**Table 1.** *Cont.*

| | USN Group (*n* = 18) | RBD Group (*n* = 10) | LBD Group (*n* = 10) | Group Difference |
|---|---|---|---|---|
| Representational drawing | 1.4 ± 1.0 * | 2.8 ± 0.4 * | | t = −5.069, df = 26, *p* < 0.001 |
| CBS (points) | 8.5 ± 5.8 | 0.0 ± 0.0 | 0.0 ± 0.0 | |
| Lesion site | | | | |
| Frontal lobe | 6 (33.3) | 1 (10.0) | 3 (30.0) | |
| Parietal lobe | 9 (66.7) | 2 (20.0) | 3 (30.0) | |
| Occipital lobe | 4 (22.2) | 0 (0.0) | 0 (0.0) | |
| Temporal lobe | 8 (44.4) | 2 (20.0) | 3 (30.0) | |
| Insular | 7 (38.9) | 5 (50.0) | 3 (30.0) | |
| Thalamus | 1 (5.6) | 5 (50.0) | 0 (0.0) | |
| Basal ganglia | 8 (44.4) | 3 (30.0) | 5 (50.0) | |
| Internal capsule | 5 (27.8) | 7 (70.0) | 2 (20.0) | |
| Superior longitudinal fasciculus | 11 (61.1) | 1 (10.0) | 1 (10.0) | |

Continuous data are presented as mean ± SD or the median (min–max) value. Categorical data are presented as n (%). Abbreviations: MMSE, Mini-Mental State Examination; BIT, Behavioral Inattention Test; CBS, Catherine Bergego Scale. *: There were significant differences between the USN and RBD groups (*p* < 0.05).

*3.2. MPT*

The results of the MPT for the USN, RBD, and LBD groups are shown in Table 2.

**Table 2.** Results of the modified Posner task (MPT) in the RBD and LBD group.

| | USN Group (*n* = 18) | RBD Group (*n* = 10) | LBD Group (*n* = 10) |
|---|---|---|---|
| Reaction time (msec) | | | |
| Valid left | | | |
| Trunk left | 993.0 (385.8–2143.3) | 400.4 ± 133.4 | 533.7 ± 89.3 |
| Trunk midline | 932.9 ± 301.7 | 495.1 ± 120.5 | 517.4 ± 119.2 |
| Trunk right | 877.0 ± 276.1 | 468.5 ± 80.3 | 502.0 ± 131.5 |
| Valid right | | | |
| Trunk left | 612.1 ± 132.1 | 446.1 ± 96.0 | 536.3 ± 97.6 |
| Trunk midline | 614.7 ± 89.6 | 447.2 ± 80.7 | 546.3 ± 169.1 |
| Trunk right | 613.5 ± 148.0 | 436.5 ± 67.2 | 528.0 ± 146.6 |
| Invalid left | | | |
| Trunk left | 1388.1 ± 651.9 | 517.1 ± 159.0 | 522.9 ± 90.4 |
| Trunk midline | 1173.0 ± 553.9 | 501.2 ± 126.6 | 521.0 ± 130.0 |
| Trunk right | 972.3 ± 351.3 | 479.9 ± 117.3 | 485.6 ± 117.2 |
| Invalid right | | | |
| Trunk left | 686.1 ± 163.3 | 431.4 ± 92.7 | 564.8 ± 117.8 |
| Trunk midline | 624.2 ± 142.0 | 465.2 ± 102.7 | 539.9 ± 140.3 |
| Trunk right | 630.0 ± 153.4 | 441.8 ± 90.0 | 522.3 ± 160.7 |
| Response rate (%) | | | |
| Valid left | | | |
| Trunk left | 93.9 (45.8–100) | 100 | 100 |
| Trunk midline | 95.1 (66.7–100) | 100 | 100 |
| Trunk right | 96.7 (75.0–100) | 100 | 100 |
| Valid right | | | |
| Trunk left | 100 | 100 | 100 |
| Trunk midline | 100 | 100 | 100 |
| Trunk right | 100 | 100 | 100 |
| Invalid left | | | |
| Trunk left | 82.4 (0.0–100) | 100 | 100 |
| Trunk midline | 88.0 (50.0–100) | 100 | 100 |
| Trunk right | 94.4 (50.0–100) | 100 | 100 |
| Invalid right | | | |
| Trunk left | 100 | 100 | 100 |
| Trunk midline | 100 | 100 | 100 |
| Trunk right | 100 | 100 | 100 |

Continuous data are presented as mean ± SD or the median (min–max) value.

For the reaction time in the trunk midline condition, a two-way repeated measures ANOVA revealed the main effects for the group ($F_{(2, 140)} = 33.41$; $p < 0.001$), target condition ($F_{(3, 140)} = 4.84$; $p = 0.003$), and interaction group × target condition ($F_{(6, 140)} = 5.24$; $p < 0.001$). In the results of the post-hoc test in USN group, the reaction time for the valid left condition was significantly delayed compared with the valid right and invalid right conditions, and the reaction time for the invalid left condition was significantly delayed compared with all other conditions. In the RBD and LBD groups, there were no significant differences between the target conditions. For between-group comparisons, the reaction times for the valid left and invalid left conditions in the USN group were significantly delayed compared with those of the RBD and LBD groups, and the reaction times for the valid right and invalid right conditions in the USN group were significantly delayed compared with those of the RBD group. There were no significant differences in the RBD and LBD groups for all target conditions.

A one-way repeated measures ANOVA or the Friedman test was used to compare the differences in each group's reaction time and response rate in the trunk midline, trunk left, and trunk right positions. In the USN group, there were significant differences in the valid left ($\chi^2_{(2)} = 7.44$; $p = 0.024$) and invalid left ($F_{(2, 34)} = 9.29$; $p = 0.001$) conditions but no significant differences in the valid right ($F_{(2, 34)} = 0.006$; $p = 0.994$) and invalid right ($F_{(2, 34)} = 2.51$; $p = 0.076$) conditions. The results of the post-hoc test showed that the reaction time for the trunk right condition was significantly reduced compared with that for the trunk left condition in the valid and invalid left conditions in the USN group. In addition, there was a significant difference in the response rate in the invalid left ($\chi^2_{(2)} = 8.90$; $p = 0.012$) condition, and the result of the post-hoc test showed that the response rate for the trunk right condition was significantly higher than that for the trunk left condition. In the RBD group, there were no significant differences in the valid left ($F_{(2, 18)} = 0.678$; $p = 0.520$), valid right ($F_{(2, 18)} = 0.181$; $p = 0.836$), invalid left ($F_{(2, 18)} = 0.510$; $p = 0.609$), or invalid right ($F_{(2, 18)} = 1.59$; $p = 0.231$) conditions. In the LBD group, there were no significant differences in the valid left ($F_{(2, 18)} = 0.814$; $p = 0.459$), valid right ($F_{(2, 18)} = 0.280$; $p = 0.759$), invalid left ($F_{(2, 18)} = 0.871$; $p = 0.435$), or invalid right ($F_{(2, 18)} = 0.862$; $p = 0.439$) conditions.

There was a significant difference in reaction time between the trunk left and trunk right conditions in the valid left, valid right, invalid left, and invalid right conditions in the USN group ($F_{(3, 51)} = 9.05$; $p < 0.001$). The results of the post-hoc test showed that the difference in the reaction time between the left and right trunk conditions in the invalid left was significantly larger than that in the valid left, valid right, and invalid right conditions.

## 4. Discussion

In this study, we performed the MPT in three different trunk positions and found that trunk position is related to attention disengagement in patients with USN. The results of this study elucidated the relationship between the temporal and spatial aspects of USN using a simple method. This finding may help explain the importance of trunk positioning in the training of patients with USN. Therapeutic intervention considering the position of the trunk based on the severity of USN may contribute to patient recovery.

In the USN group, the reaction time to stimuli in the left space was delayed in both the valid and invalid conditions of the MPT in the trunk midline condition. In addition, the reaction time for stimuli in the left space was delayed in the invalid left condition compared with that in the valid left condition. These results are consistent with those of previous studies using the Posner task [11–17]. These delayed reaction times were comparable to those of patients classified in the moderate neglect group in a previous study by Machner et al. [12]. In the invalid condition, after endogenously directing attention to the location where the target was expected to appear by referring to the cues, participants were required to find a target that appeared in a location different from that predicted. The process of detecting mismatches between the expected and actual sensory inputs requires widening attention and eye movements to new targets. This process requires switching between goal-directed (endogenous) and stimulus-driven (exogenous) attention networks, which are

independent of each other. The right temporoparietal junction (TPJ) has been implicated in detecting mismatches between expected and actual sensory inputs and switching between endogenous and exogenous attention [17,19]. However, it is known that increased memory load, such as focusing attention in the direction indicated by the cue, suppresses activity in the right TPJ [18]. Thus, it is possible that impairment of this switching between networks and increased memory load caused the delayed reaction time in the invalid right condition in patients with USN. The reaction time was delayed not only in the invalid left condition, but also in the valid condition. An endogenous orientation by the central predictive cue is less likely to cause the inhibition of return (IOR), compared with the orientation of exogenous attention. The IOR inhibits the redirection of attention to the position directed by the cue and supposedly facilitates attention to a position in the visual field that has not yet been attended [18]. Compared with the invalid left condition, the RT was lesser in the valid condition, which may imply that attention was directed to the expected location of the target and the effect of the IOR was smaller; however, this finding is still unclear, as we did not evaluate it in detail using gaze analysis.

In the present study, the MPT was performed in three trunk positions. Our results indicated that the trunk position affected the reaction time for stimuli in the left space in the valid left and invalid left conditions, especially in the invalid left condition. Reaction times for stimuli in the valid left and invalid left conditions were the fastest for trunk right condition and were prolonged for trunk midline and trunk left conditions. Patients with USN show a rightward bias in their cognitive reference for determining the center on the horizontal plane in the extrapersonal space, and the relationship between the head and the trunk position affects their ability to explore the left space [20–25]. The head-on-trunk position is related to sensory input from neck muscle proprioception, and it is known that neck muscle vibration modifies exploration behavior to the left side [24]. In other words, exploration behavior changes depending on where the stimulus appears relative to the trunk. Rorden et al. [23] investigated the relationship between trunk position and temporal factors of neglect symptoms (reaction time for stimuli). They reported that the placement of the monitor for the test on the right side relative to the trunk resulted in reduced reaction times to stimuli that appeared in left space, which is consistent with the present study. This suggests that spatial and temporal biases in patients with neglect symptoms could be related to information processing and integration. In addition, they observed a lack of relationship between trunk position and reaction time in patients with no neglect symptoms or in patients with improved neglect symptoms in the chronic phase. This suggests that spatial and temporal biases are associated only in patients with impaired brain regions related to the severity and chronicity of neglect symptoms. Our study is consistent with these results and further confirms that the reaction time to the target is influenced by trunk position. In addition, we suggest that trunk position influences switching between endogenous and exogenous attention. This implies that modulating trunk position relative to the target supports the capture of the target in the extrapersonal space based on the egocentric reference frame, even if attention is directed endogenously to another space.

The present study had certain limitations. First, only patients in whom MPT could be performed were included, and patients with severe neglect symptoms were excluded. Patients with extensive brain region damage are predicted to be more susceptible to spatial and temporal biases. Second, this study included only patients in the acute phase. Longitudinal tracking of time from the acute to chronic phase could help understand the recovery process of neglect. Third, the MMSE score of the USN group was lower than that of the RBD group, thereby suggesting this difference in cognitive function may have affected the MPT results. Future studies are required with cognitive function controlled at a similar level. Fourth, we did not perform brain imaging analyses. In addition, the percentages of brain regions summarized in Table 1 were difficult to analyze in relation to the RT because we included multiple regions for each participant. In future studies, brain imaging analysis would provide greater insights into the relationship between the spatial and temporal factors of neglect. Finally, the sample size of the present study was small. Therefore, further

data collection is needed to conduct brain imaging analysis and to examine various causal relationships according to subtypes of neglect symptoms.

## 5. Conclusions

This study revealed the relationship between trunk position and attentional disengagement using the MPT. Reaction time was reduced in the left trunk rotation condition. These results suggest that spatial and temporal biases in patients with neglect symptoms could be related to information processing and integration.

**Author Contributions:** K.S., conceptualization, methodology, validation, formal analysis, investigation, data curation, writing—original draft, writing—review and editing, visualization; K.A., conceptualization, methodology, validation, writing—review and editing, visualization, supervision, project administration; K.F., methodology, investigation, data curation, writing—review and editing; S.O., software, validation, resources; H.T., resources; S.M., resources. All authors have read and agreed to the published version of the manuscript.

**Funding:** This work was supported by funding of the Hidaka Project (03-D-1-03) in Saitama Medical University International Medical Center.

**Institutional Review Board Statement:** This study was conducted in accordance with the Declaration of Helsinki. It was approved by the Research Ethics Committee of Saitama Medical University International Medical Center (Approval No. 09-078) and Tokyo Metropolitan University (Approval No. 20043).

**Informed Consent Statement:** Written informed consent was obtained from all of the participants.

**Data Availability Statement:** The data that support the findings of this study are available from the corresponding author on reasonable request.

**Acknowledgments:** We thank the rehabilitation staff at the Saitama Medical University International Medical Center for their help during the study. We thank all participants in this study.

**Conflicts of Interest:** The authors have no conflict of interest to declare.

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
