# Peer review of "The Effect of Trunk Position on Attentional Disengagement in Unilateral Spatial Neglect"

_2035-8377, doi:10.3390/neurolint14040083_

Round 1
Reviewer 1 Report (New Reviewer)
The article is undoubtedly of interest to a wide range of researchers.
I have the following small comments on the text of the article:
1) What is t(26) in Table 1? What test did you use? In the Methods section, it is desirable to add a special subsection about statistical methods
2) In the results, it is necessary to add how the various stroke sites correlate with disturbances in the reaction time to objects in the right / left field of attention.
Author Response
Please see the attachment.

Reviewer 2 Report (New Reviewer)
Comment 1. Since the study concerns patients with left or right stroke within 30 days of onset, the authors should include in their introduction, if there are differences in the severity and the duration of neglect deficits, in the acute, sub-acute or chronic phase of stroke.
Comment 2. The authors should further explain the terms “endogenous attention” and “exogenous attention” and the differences between those two cognitive constructs. I would also suggest to add information about the relationship between neglect and attention.
Comment 3. In line 69, the authors should clarify that the integration mismatch between head, trunk and eyes leads to deficits in the visual search of left space in cases of right brain damage.
Comment 4. Since this study contains information on lesion sites and their cognitive outcomes in neglect, the authors should provide more information in their introduction about brain lesions and neglect symptoms or attention deficits.
Comment 5. The Mini Mental State Examination is a screening test of general cognitive capacity (usually administered to detect possible dementia) and not a neuropsychological tool assessing the understanding of complex instructions. Please revise the text accordingly.
Comment 6. I assume that all USN patients had a right stroke. This should be clarified in section 2.2. Moreover, the classification procedure could be described before presentation of the groups in the text.
Comment 7. Since there are patients with stroke in the occipital lobe, were these patients tested for visual deficits (e.g hemianopsia)?
Comment 8. The authors should use only one definition (reaction time or response time) throughout the text.
Comment 9. Line 188: I suggest rephrasing “statistical significance was set at α=0.05
Comment 10: MMSE score was found to be significantly inferior for the USN group, compared to the RBD group. I wouldn’t assume that this lower score may be considered as a crucial factor affecting performance of the USN group in visuospatial tasks, but the authors should at least dedicate a phrase or two to this finding (possibly a minor limitation of the study?).
Comment 11. In line 270, the term ‘inhibition of return’ needs to be clarified.
Comment 12. In line 289, I would suggest rephrasing: ‘which is consistent…’ or “a finding consistent with…”.
Comment 13. In line 304, I suggest rephrasing into’…only in patients in acute phase’.
Author Response
Please see the attachment.

This manuscript is a resubmission of an earlier submission. The following is a list of the peer review reports and author responses from that submission.
Round 1
Reviewer 1 Report
Please check the below suggestions:
1. With no clear explanation, USN is defined as "reduces or eliminates responses to the left space" in the first sentence. Right-sided USN is not acknowledged at all even though it is common although not as prevalent as left-sided USN. Please review the following articles for your references:
Esposito, E., Shekhtman, G., & Chen, P. (2021). Prevalence of spatial neglect post stroke: A systematic review. Annals of Physical and Rehabilitation Medicine, 64(5), 101459. https://doi.org/10.1016/j.rehab.2020.10.010
Yoshida, T., Mizuno, K., Miyamoto, A., Kondo, K., & Liu, M. (2022). Influence of right versus left unilateral spatial neglect on the functional recovery after rehabilitation in sub-acute stroke patients. Neuropsychological Rehabilitation, 32(5), 640-661. https://doi.org/10.1080/09602011.2020.1798255
2. The way endogenous (you need to replace the word "intrinsic") and exogenous attention system is defined is overly simplified. And how the Posner Cuing is used to differentiate them is poorly described. You need to demonstrate that you have a comprehensive understanding of endogenous vs. exogenous attention, when they can be separately under which experimental conditions (e.g., number of stimuli, SOA, cue validity) and when they are difficult to separate (such as contingent attentional capture). Please review the following articles for your references:
Chica, A. B., Bartolomeo, P., & Lupianez, J. (2013). Two cognitive and neural systems for endogenous and exogenous spatial attention. Behavioural Brain Research, 237, 107-123. https://doi.org/10.1016/j.bbr.2012.09.027
3. The BIT does assess endogenous attention but also many other visuospatial functions of the brain. A clinical test such as the BIT is a very different assessment comparing to the Posner Cuing, which was developed originally for testing stimulus-response theories in the 1980s.
4. The reason why trunk position (related to the head or facing direction) is under examination with attention disengagement is unclear. What is the theoretical reason to look into this? What knowledge will be gained if there is an association or disassociation? With no theoretical hypothesis or a priori meaningful research question, how can an experiment be designed well and conducted to collect data sufficient to answer any question? This is the main reason I do not recommend the manuscript to be published.
Reviewer 2 Report
The study investigated the role of trunk position in detecting targets in left and right space among individuals with unilateral spatial neglect, right brain damage without spatial neglect, and left-brain damage presumably without spatial neglect. They found that in spatial neglect participants, when trunk was oriented to the front while head was rotated to look at a display to the left, the reaction time was greater, compared to the same display but positioned to the right with head also rotated to the right. While this is useful to know, it’s not clear what novel contribution this study offers to our current understanding of spatial neglect mechanisms. It appears to be largely a replication study. The theoretical part of the paper suffers from the lack of clarity. A large part of the introduction is focused on brain mechanisms, but the paper neither examined nor explained how the findings impact the current neural models of spatial neglect. It would be better to spend more time on the different components of the Posner task and to explain how each comparison helps to isolate each component. Then this should be discussed in the context of cognitive theories of spatial neglect, with specific gaps identified. Lastly, the authors should point out what gap their study stands to fill. I think the paper would need to be substantially revised for publication. Also, the assumption that none of the LBD participants had spatial neglect is not valid and if there is not data to address this, it should be acknowledged as a limitation.
Introduction
Line 32: Albeit less frequently, spatial neglect occurs following left hemisphere stroke, and there are cases of ipsilesional spatial neglect. Please be mindful of that in your opening statement.
Lines 50-53: The authors state that Posner task is used to assess stimulus-driven attention and then proceed to say that it also measures goal-directed attention. Rephrase the section for consistency.
Lines 57-59: the difference between b and c is not immediately apparent. The authors should provide the contrasted conditions from the Posner task for each of those processes so that the distinction becomes clear to the reader.
Lines 63-64: Please clarify the following statement: “which requires higher attentional function than a task that responds to a target that has appeared in the neglected space”. Task doesn’t “respond”, which conditions are you referring to exactly?
Lines 65-78: The last paragraph is talking about trunk rotation. Please clarify if you mean trunk and head rotation, or trunk but not head rotation.
Materials and Methods
Line 98: Here you are talking about the BIT-conventional (a conventional subset of the original BIT tests). Typical BIT-c cutoff is <130.
Line 102: “rooming” should be “grooming”
Data Analysis
Line 159: I think you mean either post-hoc comparisons or planned contrasts (depending on whether these were pre-planned or conducted based on the achieved results). Did you mean that these tests were conducted, and that the significance level was corrected for the multiple comparisons? Then you should provide the specific correction method (Bonferroni, Tukey HSD etc).
Lines 179-182: Explain in what cases the Friedman tests were used instead of the Repeated measures ANOVA and why?
Results:
Line 185: “Based on these criteria…”, the sentence is not clear. Which criteria? You should restate the criteria or say based on the inclusion criteria.
Table 1 lists lesion site, you should explain in text how this was determined and by whom. Were all lesion sites cortical? Also, left stroke sometimes causes spatial neglect on the right. Was spatial neglect not assessed in left brain stroke participants? Was there any assessment that suggested that left brain damaged individuals did not have spatial neglect?
MPT: Why are the results for USN group shown in a figure, while the results of RBD and LBD groups shown in a table? Please follow the same formatting as in Table 1 and report the results in the table, then if you want an additional figure, you can provide it after the table. Otherwise, the reader is left wondering about the exact numeric comparisons between the spatial neglect and non-spatial neglect participants.
Figure 3. It appears that the variability of reaction times in the invalid left condition is higher than in the other conditions. Is this why the authors chose to use the nonparametric Freedman test? The exact application of ANOVA or Friedman tests must be specified, and the choice justified.
Please report how many trials were non-responses across conditions.
The statistical results for the 2 brain damaged control groups and between-group comparisons should be reported to support that the results are indeed driven by spatial neglect and not by right brain damage or by any unilateral brain damage.
Discussion
Lines 230-235: The rotation of the trunk with respect to the head, could also be called a neck rotation. Would the conclusions or implications change depending on what body part is thought to rotate? Here you are trying to dissociate head-centered and body-centered reference frames. It seems that to talk about trunk position independently of head/neck rotation, there needs to be an additional control condition where head and trunk are both rotated to the right or the left. Perhaps, a discussion about reference frames is warranted here.
In addition, sweeping statements about this finding contributing to the understanding of pathophysiology of USN and development of effective interventions need to be substantiated. How exactly our understanding is improved? What will be done differently as a result of this study to improve the effectiveness of USN interventions?
Lines 253-255: The authors are not able to make statements about the relationship between their findings and specific lesion location, as the data were not compared based on lesion location. All they can propose are hypothesized relationships that would need to be explicitly tested in future work.
Line 267: “They reported that trunk rotation to the left reduced the reaction time for stimuli that appeared in the left space, which was consistent with the present study.” This is not clear, as Rorden et al., labeled their condition where stimuli appeared on the left side of the body as “trunk left”, whereas in the current study, the authors labeled the same condition “trunk right”. It is not clear what the authors mean by trunk rotation to the left space and stimuli that appeared on the left, do the mean their definition, or Rorden et al’s? Broadly speaking the results of Rorden et al. are consistent with this study, as they showed that when the body was straight and the stimuli appeared on the left (Trunk right condition in the current study), there was a greater asymmetry in perceiving the right stimuli as appearing first compared to the left stimuli.
References
The numbered references are marked twice (e.g., 1. 1. ; 2. 2)
Reviewer 3 Report
The current manuscript studies the e relationship between trunk position and attentional disengagement in unilateral spatial neglect among acute Stroke (less than 30 days). It is important to study the factors that affect or related to the spatial neglect. Overall, the manuscript is well written and informative. Please find the following comments:
1- Mention well that study participants are in Acute stage and justify that.
2- how did you determine the participant had unilateral spatial neglect. Clarify this well in the inclusion criteria.
3- What is the clinical and research recommendations?
Round 2
Reviewer 1 Report
In the last round, I only read the Introduction because the rationale of conducting the study was inadequately described. In this round, despite of your effort revising the Introduction, it still does not demonstrate the knowledge base or the solid theory-driven reason for the study. Therefore, I did not read the rest of the manuscript, either. If you want to improve the manuscript and make significant contributions to the field, please re-write the entire Introduction section so that it provides accurate information based on what has been learned about spatial processing, attention mechanisms, and assessment development for spatial neglect. The current version does not show any of those. I cannot recommend the publication of this manuscript.
First of all, endogenous and exogenous attention are NOT mutually exclusive. It is almost impossible to separate them outside the context of experimental designs focused on covert attention. There is no "switch" between them. For example, how can the "switch" explain perseveration symptoms of spatial neglect? Thus, your account/argument that the inability to switch is the base of attentional disengagement is untrue to the current understanding. I suggest you study the following work and read more beyond them to have a good understanding about attentional disengagement. This way you can better articulate what it is and how you can examine any hypothesis related to it. (Don't just read the abstract!!)
Rastelli, F., Funes, M. J., Lupianez, J., Duret, C., & Bartolomeo, P. (2008). Left visual neglect: is the disengage deficit space- or object-based? Experimental Brain Research, 187(3), 439-446. https://doi.org/10.1007/s00221-008-1316-x
Schnider, A., Durbec, V. B., & Ptak, R. (2011). Absence of visual feedback abolishes expression of hemispatial neglect in self-guided spatial completion. Journal of Neurology Neurosurgery and Psychiatry, 82(11), 1279-1282. https://doi.org/10.1136/jnnp-2011-300656
Gandola, M., Toraldo, A., Invernizzi, P., Corrado, L., Sberna, M., Santilli, I., Bottini, G., & Paulesu, E. (2013). How many forms of perseveration? Evidence from cancellation tasks in right hemisphere patients. Neuropsychologia, 51(14), 2960-2975. https://doi.org/10.1016/j.neuropsychologia.2013.10.023
Second, the difference between ecological/functional assessment (e.g., CBS) and conventional/neuropsychological test (e.g., BIT) is not "due to the fact" that the BIT is focused on endogenous attention. There is so much discussion on the differences between the two types of assessment, and the reason you proposed is never one of them. This shows to me that you have very limited knowledge on this matter. Again, I suggest you read the following and beyond to have a much better understanding. (Don't just read the abstract!!)
Azouvi, P., Bartolomeo, P., Beis, J. M., Perennou, D., Pradat-Diehl, P., & Rousseaux, M. (2006). A battery of tests for the quantitative assessment of unilateral neglect. Restorative Neurology and Neuroscience, 24(4-6), 273-285. http://content.iospress.com/articles/restorative-neurology-and-neuroscience/rnn00351
Chen, P., Hreha, K., Fortis, P., Goedert, K. M., & Barrett, A. M. (2012). Functional assessment of spatial neglect: A review of the Catherine Bergego Scale and an introduction of the Kessler Foundation Neglect Assessment Process. Topics in Stroke Rehabilitation, 19(5), 423-435. https://doi.org/10.1310/tsr1905-423
Pitteri, M., Chen, P., Passarini, L., Albanese, S., Meneghello, F., & Barrett, A. M. (2018). Conventional and functional assessment of spatial neglect: Clinical practice suggestions. Neuropsychology, 32(7), 835-842. https://doi.org/10.1037/neu0000469
Third, because of my comments above, I cannot find a good theory-driven hypothesis for your research question. And thus, I cannot evaluate whether your study design is adequate. So I stop reading. You have to tell the readers how eye-head-trunk relative positions play a role in exogenous attention or visuospatial processing, and why it is important to know. What will it mean to the understanding of spatial neglect or spatial cognition in general?
Reviewer 2 Report
The authors substantially revised the manuscript. However, it still has significant conceptual and stylistic problems. I do think this is an interesting study with compelling set of straightforward results. Nonetheless, considerable revision is needed to make this an effective paper:
Abstract:
Please revise the last sentence by adding an explanation of how the study contributes to rehabilitation of patients with neglect symptoms (see also my point later about consistency of terminology). If using unilateral spatial neglect, use throughout, rather than using “neglect” or “spatial neglect”.
Intro:
The BIT has two subtests: behavioral and conventional. What the authors refer to as BIT is the conventional subtest. The BIT also has a behavioral subtest, that includes for example phone dialing or menu reading. The difference between CBS and BIT conventional is in that CBS measures functional deficits due to spatial neglect, whereas BIT conventional measures deficits in visuospatial processing through paper and pencil tests. However, I would argue that tasks such as visual search do not solely rely on endogenous attention but rather a combination of endogenous and exogenous attention. See for example, McLean et al., 2009; doi: 10.3758/app.71.5.1042. Compared to CBS and BIT, the Posner task is more sensitive and offers an opportunity to dissociate endogenous from exogenous processes.
Style:
Line 34: I would use RBD and LBD in please of right brain damaged vs. left brain damaged individuals.
Line 42: replace representations with “spatial representation”
Line 43 & line 46: remove “situations” after ADL
Line 45: replace “during” with “while performing”
Line 70: remove “Conversely”
Line 71: it’s not clear what the authors mean by “this integration mismatch”
Method:
Under clinical assessment (line 109), I recommend using consistent terminology when talking about spatial neglect. If using USN, please refer to it instead of “neglect”.
Line 111: here again, please specify that you are referring to the conventional subtest of the BIT. You should specify who administered assessments and what their qualifications are (e.g, trained research staff, occupational therapists, etc).
Line 126: Thank you for adding information about who identified the lesions, but you also need to explain what data were used in lesion identification (CT, MRI, radiology report).
Figure 1 appears misaligned on my copy.
Data analysis:
Lines 183-189: Here you should be running a 3-way ANOVA for midline trunk position (group x side of screen x cue validity). This will allow to partition out the interaction effect between side and cue validity. Instead, the authors are modeling condition as a simple effect, not accounting for the interaction. Similarly, for each trunk position, you shouldn’t be running one-way ANOVAs, which compounds multiple testing. Instead, you need to run a single analysis with trunk position as an additional factor. Then, you would explore the simple effects using post hoc testing.
Results
Line 207: when referring to RBD group, state “RBD group without USN”
The last paragraph on lines 254-259 indeed suggests an interaction which the authors didn’t model. Again, please separate your condition factor into two factors (side and cue validity).
Discussion
Line 263: you didn’t really study the relationship between temporal and spatial aspects of USN. Specifically, you didn’t vary timing of cues. You studied the outcome in the form or reaction time, that is not a characteristic of USN, it’s your measurement. I suggest removing this sentence.
Line 281: change “his” to “This”
Line 284: remove “also” at the start of the line
Generally speaking, try to focus your discussion to the questions raised in the introduction and the data. I would revise the discussion for clarity.
Reviewer 3 Report
Dear authors
I know well that included participants were in the acute stage. It should be clear well in methods. The motor-sensory behaviour differs in stroke stages.
The motor-sensory behaviour differs in stages of stroke. The inclusion of acute stroke has interesting points. What about the generalisability of your results. stages of stroke.
Why did not mention in the response of comments the clinical/ research recommendations. I think that you should mention and highlight in the manuscript the modifications that has been done.
Overall, the authors has not address my comments yet.